# The Miraculous Narratives in The Biographies of Eminent Nuns and The Further Biographies of Eminent Nuns

**Haoqin Zhong**

Centre of Buddhist Studies/Arts, The University of Hong Kong, Hong Kong, China; zhqhku34@connect.hku.hk

**Abstract:** This paper introduces miraculous narratives in *The Biographies of Eminent Nuns* (*BQNZ*) and *The Further Biographies of Eminent Nuns* (*XBQNZ*) and provides a comparative examination based on the relevant narratives in the above-mentioned collections and *The Biographies of Eminent Monks* (*GSZ*). First, this paper suggests that eminent nuns' miracles in the *BQNZ* seem to be more limited than those of their male contemporaries in the *GSZ*, which might reflect their comparatively limited agency in social engagements. Furthermore, the *BQNZ's* silence on the eminence of foreign nuns, in sharp contrast to the special attention afforded to foreign monks in the *GSZ*, might suggest androcentrism in both the Saṅgha and Chinese society. Second, the entries containing "intentionally performed miracles" in the *BQNZ* outnumber those in the *XBQNZ* in terms of the percentage of all entries and diversity. Moreover, in later records of the *XBQNZ*, most miracles are only related to death. This might point to the lower esteem that eminent nuns enjoyed during and after late imperial China, partly because of Buddhism's development and social status. Alternatively, this might have resulted from special social circumstances. Finally, this paper suggests that the androcentric inclination of the male compilers of the *BQNZ* and *XBQNZ*, or the sources on which the two collections are based, might have undermined eminent nuns' prominence in upholding and spreading Buddhism. Such an androcentric bias is reflected in their selective use and adaptation of the materials.

**Keywords:** eminent nuns; Chinese Buddhism; miraculous narratives; supramundane power; miraculous response; androcentrism

## 1. Introduction and Literature Review

Buddhism was introduced to China from India around the first century CE. The Buddhist monastic order was subsequently established as Buddhist Dharma spread in the country. The Chinese *bhikṣuṇī saṅgha* has continued to flourish since its founding in the fourth century. Nowadays, the Chinese *bhikṣuṇī* lineage is the main source of legitimacy for women to obtain full ordainment. Despite the rich religious heritage of Chinese *bhikṣuṇī*, however, Chinese Buddhist nuns have not received much attention from academia until recent decades, with the recognition of gender studies as an important academic subject. Female agency in East Asia has subsequently attracted considerable debate and has led to the generation of "a subfield at the intersection of cultural and historical studies, sociology, literary criticism, and material culture" (Adamek 2009a, p. 1). A growing number of studies on gender and women have been published in the context of research on Buddhism and have led to the rise of feminist Buddhology as a promising area of investigation (Barua 2011). The value of studying Chinese Buddhist nuns becomes evident when this task is situated within current research on Buddhism and feminism.

Despite the dearth of historical records on nuns, scholars have taken on the daunting job of collecting a diversity of sources, such as biographies and standard dynastic histories (Masaaki 2002; Tsai 1981), tales of miracles (Georgieva 1996), collections of poems (Li 1989), epigraphical texts (Yao 2014), inscriptions (Adamek 2009a; Pang 2010), manuscripts (Liu 2018), memorials (Adamek 2009b), etc. (Georgieva 2000; Hao 2010), to reveal the agency of Chinese Buddhist nuns over the course of history and to bring to light the previously

marginalized voices of female Chinese monastics. An interesting theme is persistent in these various sources, namely, the miraculous narratives concerning eminent nuns, not only in their biographies, or hagiographies, and miracle tales but also in standard Chinese dynastic history, epigraphical texts for nuns, and collections of poems written by them.

Miraculous narratives have the power to "compel believers to act in ways to actualize their ideal outcomes."(Weddle 2022, p. 316) Specifically, they signify the transcendent reality, confirm religious teachings, provide role models, inspire individual aspiration, and therefore play a decisive role in the development of political orders and religious institutions (Weddle 2022). The function of miraculous narratives has long been recognized by Chinese Buddhists, and such narratives permeate the relevant literature. In the first collection of the hagiographies of eminent monks, *The Biographies of Eminent Monks* (《高僧傳》 *Gaoseng Zhuan*; hereinafter "*GSZ*"), a category is reserved for *shenyi* (miracles, thaumaturges). Editors of later collections of the hagiographies of eminent monks replaced *shenyi* with *gantong*, which means "spontaneous responses from nature" (Kieschnick 1997, p. 100). According to Wang (2007), about one-third of the hagiographies in these collections contain an exemplar of thaumaturgy.

The prevalence and vital importance of miraculous narratives in Chinese Buddhism have been studied by many scholars. They believe that supramundane powers were an indispensable part of establishing the credentials of medieval Chinese Buddhist masters and creating their literary images (Sang 2007; Adamek 2009a; Yang and Christoph 2020). The sacralization of the Buddhist masters is one of the most critical features of Mahāyāna Buddhism (Chen 2022). Scholars claim that miraculous records in medieval Chinese hagiography are an important "hook" for compilers to draw attention to Buddhist doctrines, and "performed a key function in the social and economic expansion of the Buddhist Saṅgha" (Kieschnick 1997; Adamek 2009a, p. 15).

Nevertheless, the miraculous narratives of eminent Chinese nuns have attracted much less scholarly attention than those of their male counterparts. Fortunately, Georgieva (1996) introduced the relevant stories from indigenous Chinese anomalous accounts (*zhiguai xiaoshuo*) and the Buddhist literature to discuss the representation of Buddhist nuns in the Six Dynasties and the Tang period. Yet, Georgieva did not compare the miraculous stories of Buddhist nuns with those of monks and thus neglected gender-related issues in the accounting of miraculous narratives.

Recently, some scholars have claimed that the references to nuns' miraculous powers are vastly outnumbered by the accounts of monks (Adamek 2009a) and that the thaumaturgic prowess of eminent nuns in *The Biographies of Eminent Nuns* (《比丘尼傳》, *Biqiuni Zhuan*; hereinafter "*BQNZ*") is far less extraordinary than that of eminent monks in Chinese Buddhist literature (Liu 2011). However, they have omitted a detailed comparison and discussion to ground these claims. Moreover, Adamek has suggested that this phenomenon not only reflects the ubiquity of gender biases in traditional culture but also suggests that nuns made a comparatively smaller contribution to the social and economic expansion of the Chinese Buddhist Saṅgha (Adamek 2009a).

Adamek's assertion above motivates the research questions considered in this paper: First, how, and to what extent, was the miraculous power of eminent Chinese nuns eclipsed by that of their male counterparts in miraculous narratives, and what does this imply? Second, Georgieva's work has brought to prominence the miracle tales of Buddhist nuns during the Six Dynasties and the Tang Dynasty, but those after the Tang Dynasty have not been scrutinized. What is the general picture of these miraculous narratives after the Tang Dynasty (618–907 AC)? Are there any differences between these narratives and earlier ones? If so, what do they suggest? Third, do the less glorified miraculous narratives of Chinese Buddhist nuns, especially those in Buddhist hagiographies, point to a historical fact that the experiences of nuns remained "ancillary at best" in terms of upholding clerical authority and securing patronage? Is there any other viable interpretation of this?

## 2. Methodology

This paper mainly consults the miraculous narratives of the *BQNZ* and *The Further Biographies of Eminent Nuns* (《續比丘尼傳》 *Xu Biqiuni Zhuan*; hereinafter "*XBQNZ*"), two important compilations of the biographies of eminent Chinese nuns, to answer the above-mentioned research questions.

Compiled by Shi Baochang, a pious Buddhist monk in the court of the Liang Dynasty (502–557 CE), the *BQNZ* is a remarkable biographical collection that contains 65 concise biographies of eminent Buddhist nuns who lived between the Shengping period of the East Jin Dynasty (357–361 CE) and the 11th year of the Tianjian period of the Liang Dynasty (511 CE). Although it has the nature of a hagiography, the *BQNZ* provides a vivid picture of early Chinese Buddhist nuns. Moreover, it is the sole extant medieval Chinese Buddhist record on Buddhist nuns, and its value has long been recognized. Many studies on early medieval Buddhist nuns in China have taken the *BQNZ* as their starting point to discuss such issues as the ordination of women, their religious practices, relationships with politics, social status, and social activities (Tsai 1981; Yang 1997; Heirman 2010; Shi 2013; Bai 2016a).

The *BQNZ* is a rich source of accounts of miraculous narratives as well. Although Georgieva used multiple sources to report the miracle tales of nuns, she highly valued the *BQNZ* for its rich information on this topic (Georgieva 1996). Furthermore, the *BQNZ* and *GSZ* are contemporary. They were both compiled in the Liang Dynasty by monastics, and therefore share similar political, social, and cultural backgrounds. Investigating the differences between the miraculous narratives in these two collections can thus shed some light on the different ideal images of monks and nuns in view of the compilers and society at large.

Besides the *BQNZ*, there exists only one other collection of biographies of Chinese Buddhist nuns, *The Further Biographies of Eminent Nuns*, composed by the Buddhist monk Shi Zhenhua (1908–1947 CE) in the Republican times (1912–1949 CE). As a successor work of the *BQNZ*, the *XBQNZ*, in its 200 entries, also provides abundant materials on eminent Buddhist nuns from the Liang Dynasty (502–557 CE) to Republican times, covering a period of over 1000 years.

The *XBQNZ* is not contained in the Taishō collection or its supplement, and its full text is not available in the CBETA either. Nevertheless, its full text finds its way in the sixth volume of a new edition of the Chinese Buddhist Canon compiled in Taiwan[1] and the *Complete Works of Biographical Literature in Chinese Buddhism,* compiled by the Chinese National Library. Given the paucity of historical evidence on Chinese nuns, the *XBQNZ* is remarkable as it provides valuable and ready-to-use sources of information on ancient and pre-modern Chinese nuns. Yu (2020) used both the *BQNZ* and *XBQNZ* as main sources to explore the development of Chinese *bhikṣuṇī saṅgha* in the history. Tho (2008), based on twelve biographies in the *XBQNZ*, examined Chinese Buddhist nuns' religious practices and influence in ancient and pre-modern China and their position in pre-modern China. In addition, some articles have referred to the *XBQNZ* to discuss Chinese Buddhist views towards women during different periods (Liu 2021) and the general subject of ancient Chinese Buddhist nuns, for example, their observation of garudharma (Wang 2011) and their translation ability (Zhuo 2012).

Miraculous narratives are also ubiquitous in the *XBQNZ*. Though in the Republican times, the development of modern science and society galvanized many Buddhists to take collective actions to modernize Buddhism, the pious Buddhists never questioned the ultimate religious goal of Buddhism—enlightenment (Ashiwa and Wank 2009, pp. 52–55). Zhenhua, as one of those pious Buddhists, did not seem to have questioned the achievements of earlier eminent practitioners either, in which miraculous power and responses were included. Huang (2016) introduced the idea that Zhenhua fully believed that the miraculous narratives in the novel *Master Huangxin* were based on solid historical facts instead of being fabricated. Therefore, he wrote to the author several times, asking for more historical materials on eminent nuns for his compilation of the *XBQNZ*. Seen from this, Zhenhua did not have a negative attitude towards miraculous narratives while producing

the compilation. Of course, Zhenhua might have had limited sources for the compilation; therefore, it was possible that he did not include all the miraculous narratives of eminent nuns in the history. However, he should not have missed these narratives intentionally.

By introducing the miraculous narratives in the *XBQNZ* in comparison with those in the *BQNZ*, this paper aims to present and interpret miraculous narratives of eminent nuns after the Liang Dynasty in the *XBQNZ*.

Moreover, to answer the third research question mentioned above, this paper refers to sources other than the *BQNZ* and the *XBQNZ*, such as standard dynastic history, miracle tales, and epigraphical texts, to confirm the information provided by these two biographical collections. In so doing, we draw a balanced portrayal of eminent nuns in Chinese history and provide new insights into female agency in Chinese Buddhist monasticism.

### 3. Different Kinds of Miraculous Narratives in the *BQNZ*

An ancient and fixed feature of Buddhism is that a certain specific "higher (super-) knowledge" (*abhiññā*): is integrated into the process of liberation. The *Puggalapaññatti* (pp. 184–85) explains that all arahants enjoy two-fold liberation: liberation of the mind and liberation by wisdom. The *ubhatobhagavimutta* arahant (one liberated in both ways) can additionally experience liberation from the material body (*rūpakāya*), which brings the practitioner six kinds of super-knowledge (*abhiññā*)

1.  Psychic power (*iddhividhā*). The *Visuddhimagga* (pp. 373–405) further explains *iddhividhā* as multiplying oneself; going unhindered through walls, enclosures, and mountains; diving out of and into the river; traveling while seated cross-legged; touching the moon and the sun; transformation; and making a mind-made body.
2.  Knowledge of the divine ear (*dibbasotadhātu*). This represents the ability to hear sounds that are divine and human, and distant as well as close (*Dīgha Nikāya* I, p. 79).
3.  Knowledge of penetration of minds (*cetopariyañāna*).
4.  Knowledge of recollection of past lives (*pubbenivāsānussati*).
5.  Knowledge of the death and rebirth of beings (*dibba-cakkhu-cutūpapātañāṇa*). This is the ability to know the degeneration and ascent of beings according to their karma as low existence, excellent existence, good conditions, bad conditions, good destinies, and bad destinies (*Dīgha Nikāya* I, pp. 82–83).
6.  Knowledge of final liberation (*āsavakkhayañāṇa*)[2].

*Chaḷabhiññā* plays an essential role in the miraculous narratives of early Buddhism. However, it is insufficient to relate the complexity of the miracles performed by Buddhas and Mahābodhisattvas in Mahāyāna literature. In China, the complex of supernormal power and miracles developed and expanded following Kumārajīva's translation of *Mahāprajñāpāramitā Śāstra* all the way to Xuanzang's translation of the *Yogācāra-bhūmi*, which provided a more "robust and complex" categorization of miracles performed by Buddhas and Mahābodhisattvas (Shi 2022, p. 92). This development occurred partly because Mahāyāna places a greater emphasis on the Buddhas' and Bodhisattvas' supreme power as saviors for the faithful. Stories concerning the miraculous response of the Buddhas and Bodhisattvas became an effective way to inspire and sustain the faith of the devotees (Yu 2007).

Accordingly, miraculous narratives in Chinese hagiographical collections contain two themes. One is that the eminent monastic practitioners themselves had *chaḷabhiññā*, and performed miracles intentionally. The other is the testimony of the efficacy of the supernormal power of the Buddhas, Mahābodhisattvas, and Buddhist teachings, with virtuous eminent monks and nuns as intermediaries.

However, compilers have not separated the above two themes since the beginning of the biographical tradition and have always classified them into one category—either as *shenyi* by Huijiao or as *gantong* by later compilers. To provide a neater introduction to the miraculous narratives and a clearer comparison between the three collections mentioned above, this paper refers to the scheme of Georgieva (1996) and divides the narratives into two categories. The first consists of miracles that were intentionally performed by eminent

nuns, which emphasizes the spiritual powers of the main characters. The second consists of miraculous responses, either to save or protect the faithful and virtuous eminent nuns, or as "unexpected events"—beyond the expectation of common people—that result from the genuine faith and pure practices of eminent nuns[3]. The second category puts more emphasis on the main characters' stimulation of the miraculous responses.

Two issues need to be pointed out here. First, if the protagonist (or protagonists) in a single entry in the text experienced both kinds of miracles above, the entry is referred to in both categories. Second, the miraculous narratives discussed here should involve a nun playing the main role. The narratives in which a nun merely witnesses other people perform miracles are not included.

Of a total of 65 entries, 31 entries in the *BQNZ* contain accounts of miraculous narratives.

### 3.1. Intentionally Performed Miracles

(1)　*Iddhividhā*

The best example is the nun Daorong, who converted Emperor Jianwen (r. 371–372 CE) from Taoism to Buddhism by her effective *iddhi*.

During the rule of Emperor Ming of the Jin Dynasty (r. 323–326 CE), Daorong manifested her *iddhi* by making the flowers under her sitting cushion blossom and garnered significant respect from the emperor.

Emperor Jianwen was initially a Taoist and refused instructions on Buddhism from Daorong. Subsequently, he saw spirits (ghosts) whenever he entered the Taoist hall, all taking the shape of a *śramaṇa*. The (future) emperor suspected that Daorong had caused this, but nobody could explain how. Later, after he had ascended to the throne, there was an evil omen in the palace: crows nestling in the Taiji hall. Daorong asked him to fast for 7 days, and to receive and uphold the eight precepts of Buddhism to disperse the crows. Her advice proved effective and inspired great faith in the emperor. He subsequently treated Daorong as his master[4].

Also in the Jin Dynasty, Jingjian reportedly rose in the air and walked up to the sky on a rainbow-like road when she knew that it was time to leave this world[5].

In the Qi Dynasty, Fayuan suddenly disappeared for 3 days and subsequently claimed that she had been to Tuṣita heaven to listen to the teachings of the Buddha. Later, she again disappeared for 10 days and returned with the ability to read foreign languages and recite the sūtras[6]. Zhisheng rubbed incense and made it burn, and Dele could clearly see in a dark room[7].

(2)　*Dibba-cakkhu-cutūpapātañāṇa*

Jingxiu of the Liang Dynasty bade farewell to her monastic acquaintances 5 days before her death and predicted that she would go to the heaven of Tuṣita[8].

(3)　Advanced Stages of Meditation

In the Liu Song Dynasty, Sengguo meditated for several days with her body cold and her flesh stiff. When people wanted to move her body, she emerged from the state of meditation and talked and smiled as usual[9]. Faxiang meditated for 3 days without moving, with her body still as a rock or a piece of wood, and Fabian meditated similarly for some time[10].

Moreover, Daozong and Huiyao kept reciting sūtras while self-immolating. Even when the fire reached their faces, eyes, and ears, they were not affected[11]. A similar incident occurred with Shanmiao. When the fire rose to the top of her head, she kept calm, summoned the other nuns, and bid farewell to them[12].

(4)　The Ability to Make Predictions

In the Liang Dynasty, the nun Feng predicted that the monk Fahui would attain the third supramundane fruit (*anāgāmī*: non-returner) after listening to the preaching of Mas-

ter Zhiyue in Kucina. Later, Feng again predicted Fahui's return and waited with her disciples to greet him[13].

*3.2. Miraculous Responses*

(1) Related to death

The miraculous aspects related to death include the nuns' perception of their approaching death, dreams suggesting it, auspicious omens such as fragrance and light, and the strange behaviors of animals after their death.

In the Jin Dynasty, when Jingjian saw a woman with colorful flowers, she knew that it was her time to leave this world[14]. Lingzong dreamed of Sumeru before death[15], suggesting that she would pass away soon. In the Liu Song Dynasty, just before her death, Fasheng dreamed that the Buddha discussed the two *yānas* with two Mahābodhisattvas, and that they all came to visit her to ask about her illness[16]. Fasheng knew that her death was imminent after she saw two strange monks and the Buddha sitting on a lotus at a distance and shining a special light on her body[17]. Puzhao dreamed of a tower with a monk in it 7 days before she passed away[18].

In the Qi Dynasty, Zhisheng announced her death 3 days before it occurred[19]. In the Liang Dynasty, Jingxiu dreamed of Buddhist flags and umbrellas (*patākā, chatta*) and musical instrumentals to the west of the Mahavira Hall days before her death[20].

A special fragrance and red light were noted before Jingjian's death, and she saw a woman with colorful flowers[21]. A special fragrance and an auspicious omen filled the sky when Guangjing of the Liu Song Dynasty passed away[22].

Also in the Liu Song Dynasty, Huiqiong's corpse did not decay for more than 10 days after her death. Moreover, birds and beasts did not look to pick at her corpse and did not even peck the rice beside her body[23].

(2) Miraculous help

In the Jin Dynasty, Lingzong was being chased by thieves when a white deer appeared and led her across the river[24]. In the Liu Song Dynasty, the parents of Sengduan refused to let her enter monasticism and forced her to marry. As she recited the sūtra continuously with piety, the Buddha appeared to inform her that her husband-to-be would be killed by a bull the next day. It happened accordingly, and she finally went forth[25].

(3) Taming animals

In the Jin Dynasty, the residence of Zhixian was surrounded by birds who followed her wherever she went[26]. In the Liu Song Dynasty, a tiger was always following Jingcheng[27].

(4) Magic cure and other omens related to going forth

In the Liu Song Dynasty, before going forth, Daoshou fell ill with a severe disease. She swore that if she recovered, she would go forth. She did recover, and saw a Buddhist umbrella (*chatta*) covering her body in a dream. Then, she went forth according to her oath. Similarly, before going forth, Xuanzao prayed for 7 days to the Avalokiteśvara and asked for recovery from severe disease. Her prayer worked, and she saw the power of Buddhism and went forth[28].

In the Qi Dynasty, when Sengjing was born, her mother was told by a voice in the sky to let the baby girl go forth. The mother adhered to this instruction[29].

In the Liang Dynasty, the (future) nun Faxuan was always covered by a curtain when she was sleeping and sitting. After her ordination, the curtain disappeared forever[30].

(5) Sympathetic resonance (*ganying*) in meditation

In the Liang Dynasty, Tanhui saw two beams of light while meditating. She recognized that the white beam indicated the Bodhisattva path while the green one indicated the *śrāvaka* path. Huisheng could perceive various miraculous visions while meditating and interpret them[31].

(6)    Auspicious phenomena

In the Jin Dynasty, whenever Jingjian engaged in *karma*, a strong, special fragrance hung around her. Whenever Minggan performed a repentance ceremony, she did not stop until some auspicious phenomena, such as raining flowers, a voice in the sky, the appearance of the Buddha, or good dreams, occurred[32].

In the Liu Song Dynasty, after Huiyu had performed ascetic practice for 7 days, a bright light shone in her temple as testimony that she would see the Buddha after death. Golden light was emitted from between the eyebrows of the Buddha statues set up by Daoyuan. The temple of Jingcheng was seen at a distance to glow with light at night[33]. When Huimu was performing a repentance ceremony, she saw the monastic altar and the sky turn golden. Then, she saw a man who informed her that she could take Buddhist precepts. On the night before she received the precepts, they were taught to her in a dream[34].

In the Liang Dynasty, there was always a voice in the sky reminding other nuns not to disturb Jingxiu. Jingxiu's transcriptions of the sūtras prompted two brothers of the dragon kings (*sāgara*) to appear to show their appreciation and support for her. Moreover, every time Jingxiu offered food to eminent monks, auspicious phenomena were reported.

*3.3. Discussion*

Miraculous tales were considered to be factual events and were even included in standard historical accounts in medieval China. Buddhist miracle stories were thus important for convincing Chinese people of the magical efficacy of Buddhist teachings by providing live evidence. Therefore, the Saṅgha's ability to wield supernormal power or access divine power as a powerful intermediary is a ubiquitous theme in Buddhist hagiographies.

Nevertheless, although both served as intermediaries to demonstrate the power of Buddhism, the portrayals of nuns in these narratives are different from those of monks in two prominent ways.

First, all the eminent nuns in the miraculous narratives—and all of them in the *BQNZ*—are native Chinese, while many eminent monks—35 in all—are said to have come from foreign lands, such as India, Sogdianna, and Kucha, constituting 14% of all the entries. Moreover, among the 35 foreign monks, 20 demonstrated astonishing miraculous powers, constituting 57% of all. Their foreign origins gave them "an aura of exoticism and mysteriousness," and lent greater credibility to their miraculous narratives (Salguero 2010, p. 17).

However, no miraculous narrative—or any narrative actually—in the *BQNZ* is associated with foreign nuns. From Sengguo's and Baoxian's entries in the *BQNZ*[35] and also Saṅghavarman's entry in the *GSZ*[36], we know that Sinhalese nuns came to China and also conferred "the second ordination" on Chinese nuns. These Sinhalese nuns manifested their eminence in at least two ways: their determination to spread the Dharma and their knowledge of the Vinaya. Yet, the compiler, Baochang, did not dedicate an entry to them, nor are they included in the miraculous Chinese narratives in indigenous repertories.

The invisibility of foreign nuns, in sharp contrast to the special attention afforded to foreign monks, reflected foreign nuns' marginalized status compared with their male counterparts. The androcentrism in both the Saṅgha and Chinese society might have been one reason for this. Of course, the social backgrounds of the Sinhalese nuns and the Chinese nuns who got ordained by them, or their social activities, might also have contributed to such negligence.

Second, although the Six Dynasties are viewed as the heyday for early Chinese nuns, and eminent nuns received high respect in these times, their miracles as reported in the *BQNZ* appear limited compared with those of their contemporary male counterparts in the *GSZ*. Salguero has noted that in the *BQNZ*, nuns do not manifest the range of thaumaturgical and transformational abilities that monks from the *GSZ* exhibit in terms of healing others. He suggests that this may indicate either a bias against healing by females and the gendered conception of the ability to wield supermundane powers or simply "a difference in the interests of the compilers" (Salguero 2010, p. 17).

Eminent nuns are absent not only from accounts of miraculous healing but also from many other areas. Wright (1990, p. 38) summarized how Fotucheng demonstrated the power of Buddhism in four fields:

(1)  Agriculture. His prowess as a rainmaker ensured the stability of a predominantly agricultural state.
(2)  Warfare. He advised on successful war plans based on his supramundane capability of prediction.
(3)  Healing. He acted as a savior for the depopulated and epidemic-ridden country.
(4)  Politics. With his abilities of prediction and mind reading, he helped the ruler maintain a balance of political power.

Fotucheng can be viewed as a typical eminent monk in the *GSZ* who served as an imperial advisor. However, none of the above four powers was included in the miraculous narratives in the *BQNZ*. Most miraculous narratives of eminent nuns are related to their personal experiences and Buddhist practices. Daorong is the only one whose miraculous power is mentioned in association with the emperor. However, she did not participate in political issues at all. Therefore, these records indicate that even eminent nuns in the Six Dynasties did not enter politics. Accordingly, their agency to spread Dharma might also have been limited.

**4. Miraculous Narratives in the *XBQNZ* in Comparison with Those in the *BQNZ***

There are 85 entries containing miraculous narratives in the *XBQNZ*.

*4.1. Intentionally Performed Miracles*

(1)  *Iddhividhā*

The nun Fayuan's *iddhi* of traveling at will found its counterpart in the record of Jingzhen of the Tang Dynasty in the *XBQNZ*. She traveled around making offerings to the Buddhas in her meditation (Shi 1983, p. 6).

The nun Zhisheng's ability to command fire is mirrored by the ability of Xinxiang of the Tang Dynasty to control water. When Xinxiang was performing water-element meditation, her figure disappeared and people just saw water in her meditating room (Ibid. p. 5).

Other *iddhi* in the *BQNZ* have no counterparts in the *XBQNZ*.

(2)  *Dibba-cakkhu-cutūpapātañāṇa*

The nuns Wuxing and Fakong of the Tang Dynasty had *dibba-cakkhu-cutūpapātañāṇa*, just as Jingxiu did. Before passing away, Wuxing told her acquaintances that she would be reborn in the Pure Land and that her peer, who had been reciting the name of the Buddha diligently, would also be reborn there in due time as she saw that the lotus had been prepared for them (inside which to be reborn) (Ibid. p. 17). Fakong also predicted her rebirth in the Pure Land before passing away (Ibid. pp. 19–20).

(3)  Advanced stages of meditation

The accounts of the abilities of the nuns Sengguo and Faxiang to attain *dhyāna*, with their bodies cold and stiff, have no counterparts in the *XBQNZ*.

The maintenance of a strong concentration during self-immolation, as reported in the *BQNZ*, is echoed by the account of two nuns from Jingzhou in the Tang Dynasty, who kept reciting the sūtra even when the fire burned their eyes (Ibid. pp. 5–6) and the old nun on the lake in the Qing Dynasty, who kept calling the name of the Buddha even when her body had been scorched (Ibid. p. 96).

(4)  The ability to predict

The ability of the nun Feng in the *BQNZ* to tell the future is echoed in the *XBQNZ* in the accounts of Zhixian of the Sui Dynasty, Gongdeshan of the Tang Dynasty, and two nuns from the Ming Dynasty: Changjing and Jixing (Ibid. pp. 48–49).

Zhixian predicted that Wendi of the Sui Dynasty (r. 581–604 CE) would be the emperor just after his birth. She also predicted Wendi's revival of Buddhism (Ibid. pp. 3–4). Gongdeshan predicted the timing of Daizong's (r. 762–779) return to the palace after the Rebellion of An and Shi: when he saw a cow (Ibid. p. 17). These two nuns' predictions were related to state politics, while those of the two nuns in the Ming Dynasty were related to ordinary fortune telling for the public.

(5)    *Pubbenivāsānussati*

The *XBQNZ* touches on knowledge of the recollection of past lives, which does not appear in the *BQNZ*.

Sengfa of the Liang Dynasty uttered 21 sūtras when she was only around 8 or 9 years old (Ibid. pp. 1–2). The author commented that, although some claimed that she had been taught by deities, they would rather accept the idea proposed by the Buddhist scholar Fei: that she had learned this in one of her past lives and had recollected it thence. Receiving teaching from deities accords with the Taoist concept of spirit writing, while *Pubbenivāsānussati* is its Buddhist interpretation.

In the Song Dynasty, Huangxin received sudden enlightenment and remembered experiences from her past life, when an ugly nun shouted to her. This is a typical Zen story (Ibid. pp. 37–38).

Suwen of the Qing Dynasty knew the karmic relationship between her and her parents: she had been a monk in her past life and received offerings from the previous incarnation of her parents (Ibid. pp. 107–8).

*4.2. Miraculous Responses*

(1)    Miracles related to death

Originating in the *BQNZ* in the context considered here, this subject seems to be a perpetual theme in the *XBQNZ*. Miraculous dreams portending death, a special fragrance and light upon death, abnormally preserved corpses, and special respect from animals in the *BQNZ* all find counterparts in the *XBQNZ*. Moreover, these miracles are further developed in the latter text.

First, the deaths of eminent nuns inspired many auspicious omens according to the *XBQNZ*, such as purple clouds (for Miaokong of the Tang Dynasty (Ibid. pp. 4–5)), music (for Wuxing of the Tang Dynasty and Nengfeng of the Song Dynasty (Ibid. pp. 17, 38)), and the appearance of lotus flowers (for Hongyuan in the era of the Republic (Ibid. pp. 123–24)). Moreover, reports of a special fragrance have been mentioned in relation to the deaths of nine eminent nuns and the fragrance of lotus flowers in relation to the passing of three.

Many other unusual events have been reported: for instance, the nose of a nun, Dumu jingang (Vajradhara with one eye only), in the Ming Dynasty catching fire and the rice bowl in the hand of Shengdao, in the time of the Republic, spinning like a flower and rising to the height of a person. It spun for longer than 15 min and then descended on the steamer without losing a single grain of rice in the process (Ibid. pp. 54, 121–22).

Second, many eminent nuns perceived their impending death by sympathetic resonance in various ways. Some were instructed by the Buddha, for example, Decheng from the time of the Republic (Ibid. pp. 125–26). Others were instructed by Mahābodhisattvas, such as Fakong of the Tang Dynasty and Nengkai from the era of the Republic by Mañjuśrī (Ibid. pp. 17, 38), Zhizang of the Song Dynasty by Kṣitigarbha (Ibid. p. 26), and Chengci of the Ming Dynasty by Maitreya (Ibid. pp. 55–56).

Still others were enlightened by special dreams. For example, Langran of the Qing Dynasty dreamt of herself sitting crossed-legged on the (lotus) flower in the Pure Land three times (Ibid. p. 99). In the time of the Republic, Xinzhong dreamt that a vehicle with Buddhist marks wanted to take her away (Ibid. pp. 129–30). The meaning of the dream was evident. Guoren had two miraculous dreams indicating her death. In one, four boys carrying flags and a sedan chair claimed to be taking her to the Western world. In another, a monk related to her the exact date of her final departure (Ibid. pp. 119–20). Shengdao

also had two miraculous dreams. In one, a person announced that she would be taken to the Western world soon. In another, she met an old woman in Vulture Peak who claimed that she would take the nun to the Western world soon (Ibid. pp. 121–22).

While other eminent nuns also reportedly perceived their deaths in advance, how they realized this has not been detailed. The following lists them:

In the Tang Dynasty: Facheng and Huiyuan (Ibid. pp. 10–13).
In the Song Dynasty: Nengfeng and Liaozheng (Ibid. pp. 19–21, 124).
In the Ming Dynasty: Chengjing (Ibid. pp. 56–57).
In the Qing Dynasty: Xinggang, Chaoyin, Xingxuan, Foyin, Chaochen, Benyin, Suiqin, Lvzong, Foqi, Miaocheng, Daogan, Daowu, Mingheng, and Lingyi (Ibid. pp. 60–62, 68, 72–73, 77–78, 96–102, 106, 111).
In the Republic era: Lianzhen, Yingen, and Yinxin (Ibid. pp. 116–17, 122–23, 129–30).

Third, eminent nuns themselves saw the Buddha or Bodhisattvas welcoming them before their deaths. Ruzhan of the Song Dynasty, Miaocheng and Daowu of the Qing Dynasty, and Dawu from the time of the Republic saw the Buddha leading their way just before they passed away (Ibid. pp. 40, 99–100, 102, 120–21). Chengjing of the Ming Dynasty and Ruzhi from the time of the Republic saw Avalokiteśvara in their last moments (Ibid. pp. 56–57, 113–14). In addition, Foqi of the Qing Dynasty saw Mahābodhisattva 3 days before her death, and Hongyuan of the time of the Republic saw Maitreya several times a few days before her death (Ibid. pp. 97–98, 123–24).

Fourth, mysterious occurrences have been reported in the *XBQNZ* regarding the corpses and coffins of eminent nuns that are not found in the *BQNZ*. For example, a flower arose in the mouth of the corpse of Daoji, a Liang Dynasty nun, the corpses of three nuns from the Tang Dynasty had intact tongues after self-immolation (including two nuns in Jingzho), and that of one corpse that had been disposed of in the forest had also remained pristine (Farun) (Ibid. pp. 1, 5, 24). This may be a testimony of their extraordinary ability to speak true Dharma.

The corpse of Demi from the Qing Dynasty, with a jade under her nose and light on her head (Ibid. p. 63)—just like the imperishable corpse of the eminent nun in the *BQNZ*—is a testimony to her virtues and achievement. The same applies to the nun in the Lanruo nunnery and Benlian, whose corpses did not decay for days in the hot summer (Ibid. pp. 101–2, 109–10).

Records from the time of the Republic add one aspect to the corpses of the eminent nuns: a warm vertex after death (Ruzhi, Gaofeng, Rujue and Hongyuan) (Ibid. pp. 113–16, 123–24), as evidence of their rebirth in the Pure Land. This has not been found in the previous records in the *XBQNZ* and those in the *BQNZ*, which suggests that this concept was developed during this period.

Moreover, the mysterious movement of the coffin of Yuanji from the Tang Dynasty suggests her spiritual achievement (Ibid. pp. 15–16).

Fifth, many eminent nuns mentioned since the Tang Dynasty are reported to have passed away in the sitting position: one during the Tang Dynasty, seven during the Song Dynasty (two who had been reciting the sūtra), four in the Ming Dynasty (one reciting the sūtra), twenty during the Qing Dynasty (seven reciting the sūtra, or calling the name of the Buddha), and three in the time of the Republic (all of whom had been reciting the sūtra). The sitting position seems to have been a standard posture for eminent nuns who passed away, as reported in later periods. This seems to reflect the prevalence of the Pure Land tradition in these periods. Nevertheless, some strange postures have also been mentioned: for example, a nun during the Song Dynasty (Hui'an) passed away in an upside-down standing position (Ibid. p. 39).

In addition, cases have been reported of getting *śarīra dhātu* (*sheli*, special bodily relics) after the cremation of eminent nuns since the Tang Dynasty, such as the nun Fakong of the Tang Dynasty, Pugui of the Yuan Dynasty, Wuwei of the Ming Dynasty, and Tongdao, Dawu, and Decheng from the time of the Republic (Ibid. pp. 19–21, 47, 51, 112–113, 120–21, 125–26).

Since reports from the Song Dynasty, eminent nuns have been recorded leaving final verses, but not many—in all, two in the Song Dynasty, one in the Ming Dynasty, and six in the Qing Dynasty. Moreover, none of the eminent nuns left final verses during the time of the Republic. This might have reflected the development of the Zen tradition.

(2)　　Sympathetic resonance in meditation

The experience of perceiving miraculous visions during meditation, as reported for Tanhui and Huisheng in the *BQNZ*, finds counterparts in the experience of Chengci of the Ming Dynasty and Decheng of the time of the Republic in the *XBQNZ*.

In her meditation, Chengci was led by a Bodhisattva to visit the Pure Land and meet Maitreya (Ibid. pp. 55–56). In 1932, when Decheng was going through a 21-day retreat, she engaged in calling the name of the Buddha. Then, she saw four Chinese characters (selfness, mercy, permanence, nirvana) in her concentration and heard that the Buddha had called her name[37].

(3)　　Receiving miraculous help or instruction

According to the *BQNZ*, Lingzong was helped by a deer with magical powers when chased by thieves, and Sengduan was assisted in entering the Saṅgha after the mysterious death of her husband to be. These were the practical difficulties ahead of the nuns or nuns to be in the Six Dynasties.

Nevertheless, these difficulties—especially the latter—were not prevalent in later dynasties. The major hardships that nuns in the Tang Dynasty faced were the sexual advances of laymen (nun Hunshan) and encroachments on property by greedy monks (nun Faxin) (Ibid. pp. 4, 9). These two issues were also related to miraculous help in the entries of nuns at that time in the *XBQNZ*. The nun Hunshan recited the *Lotus Sūtra* twice a day for more than 20 years. Her virtues reportedly caused the man who wanted to violate her to lose his male organ. The special respect afforded by Faxin to the manuscript of the *Lotus Sūtra* rendered it unopenable by the greedy monk who wanted to take the manuscript from her.

In addition to respect toward the Buddhist sūtra, respect toward portraits of Mahābodhisattva yielded miraculous help. In the Song Dynasty, strong winds destroyed the house of Zhizang and blew away portraits of the Western Trinity, Kṣitigarbha, and Avalokiteśvara, which she worshiped. She prayed to Kṣitigarbha and the portraits flew back to her, glowing like lightning (Ibid. p. 26).

(4)　　Taming animals and protection by deities

The theme of taming animals and protection by deities is echoed in accounts of Daoji of the Liang Dynasty and Facheng of the Tang Dynasty in the *XBQNZ*. When Daoji was explaining the Dharma, white birds[38] would join her audience. When Facheng practiced ascetics in the forest, leopards always followed her and the deities protected her (Ibid. pp. 1, 10–11).

(5)　　Magical cures and other omens related to going forth

In the *BQNZ*, a magical cure always enabled women to go forth. Two entries relate to special omens predicting their going forth. However, such narratives are absent from the *XBQNZ*.

Buddhism was still in its formative stage in the Six Dynasties and encountered criticisms from society. Therefore, the eminent nuns might have needed a miraculous narrative to sanction their choice of going forth. Once Buddhism had taken root in China in later dynasties, the hagiographies might not have needed to explain their decision any longer. Therefore, the only magical cure in the *XBQNZ* testified to the virtues of a Ming Dynasty nun, Dumu jingang (Ibid. p. 54).

(6)　　Other auspicious phenomena and unexpected miraculous experiences

The omens during the *karma* ceremonies, or the miraculous teaching of the precepts, are not found in the *XBQNZ*. Nevertheless, there are other themes in it with no counterparts in the *BQNZ*.

First, with regard to auspicious omens, a nun in the Chen Dynasty was diligent in reciting the *Lotus Sūtra*. When she was once reciting the *sūtra*, flowers grew on the five fingers of her right hand and the left palm in order (Ibid. p. 2).

Second, there are accounts of the efficacy of the virtues of eminent nuns, like selfless sacrifice, ascetic practices (other than self-immolation), piety, and mastery of the Dharma. In the Song Dynasty, Huangxin's nunnery failed eight times to build a large copper clock. On the ninth attempt, Huangxin, as the abbess, sacrificed her life by jumping into the furnace. This attempt subsequently succeeded owing to her selfless sacrifice and supreme faith (Ibid. pp. 37–38).

Similar selflessness is reflected in the entry on Suwen of the Qing Dynasty. When her mother perceived relics of the Buddha in the tower as black, Suwen knew it was because of her mother's bad karma. Therefore, she burned her finger in front of the tower to offer repentance for her mother. The next day, when her mother watched the relics again, she saw one as a larger gold bead and one as a smaller white bead, which meant that her bad karma had been redeemed. Moreover, Suwen herself perceived the relics as pure, bright crystals, which was a testimony to her pure mind (Ibid. pp. 107–8).

Moreover, in the Song Dynasty, nun Daohui, who was versed in the Dharma, invited rain during drought by pious pray (Ibid. p. 41). In the Ming Dynasty, Huixiu found a fountain to help the villagers. This was believed by her contemporary to be a testimony to her rigorous ascetic practices (Ibid. p. 53).

Third, there are accounts of eminent nuns' interactions with or visions of Mahābodhisattvas or the deities.

The nun Fakong of the Tang Dynasty, who was then leading a wandering life, was instructed by Mañjuśrī. The latter, taking the form of an elder, told her to settle down on Mount Wutai and predicted that she would achieve the Dharma fruit in so doing. Also in the Tang Dynasty, Chifa and her sister Huiren were visited by an illuminated nun named Aunt Kong, who was said to be an incarnation of Samantabhadra. Zhenru, from the Tang Dynasty, passed the magical sachets given by the deity to Emperor Daizong, which helped him quell the Rebellion of An and Shi (Ibid. pp. 19–21, 11, 16–17).

Two special miraculous narratives from the Qing Dynasty in the *XBQNZ* reflect a prevailing famine. Dengling, after eating some mysterious red pellets, acquired the divine ability to stay satiated without eating ordinary food. The nun Dong used offerings to feed beggars following the instruction of Avalokiteśvara in a dream (Ibid. pp. 59–60, 92).

Special visions owing to true faith have been related in the following narratives: in the Song Dynasty, Nengfeng always saw Buddhist light shining on her, and Ruzhan always saw Buddhist light in the room. In the Qing Dynasty, Qingyue kept reciting the *Sukhāvatīvyūha Sūtra* on her way to Mount Wutai to make offerings to 500 monks and saw 6 mysterious monks who later flew into the forest. No one else could see them, however. In the time of the Republic, Tongdao saw the Western Trinity on a roof during her practice (Ibid. pp. 38, 40, 100, 112–13).

Finally, there is also a unique narrative focusing on the afterlife of an eminent nun. The corpse of the nun Benlian could not be destroyed by fire, and people subsequently decorated it with gold to worship it. They then put a wooden crown on its head. Later, an old woman who used to be close to Benlian had a dream in which Benlian complained to her that the crown was so heavy that it pressed against her head. Benlian asked her to remove it. The old woman related the dream to the villagers, who found that this was indeed the case. They removed the wooden crown and put a paper crown on her head only in the winter (Ibid. pp. 109–10).

In the time of the Republic, Bendao worshiped statues of the Buddha (or Mahābodhisattvas) twice a day for several decades. She left footprints several inches deep where she always stood and worshipped (Ibid. pp. 145–46). This was viewed as miraculous by the locals.

### 4.3. Discussion

The 85 entries containing miraculous narratives in the *XBQNZ* constitute 43% of the total, which is slightly less than the 48% of entries in the *BQNZ* that report such miracles. Nevertheless, there is a significantly smaller percentage of entries containing "intentionally performed miracles" in the *XBQNZ*. The table below shows a detailed comparison.

As shown in Table 1, the percentage of entries containing intentionally performed miracles significantly declined from the Tang to the Song Dynasty. They underwent a resurgence in the Ming Dynasty but then drastically decreased, with no entries from the time of the Republic.

**Table 1.** Entries containing intentionally performed miracles.

| | BQNZ | XBQNZ | | | | | | | | |
|---|---|---|---|---|---|---|---|---|---|---|
| | | Six Dynasties | Sui | Tang and Five Dynasties | Song | Yuan | Ming | Qing | The Republic | Total |
| Number of all entries | 65 | 5 | 1 | 30 | 29 | 8 | 14 | 86 | 27 | 200 |
| Entries containing miraculous narratives | 31 | 3 | 1 | 15 | 12 | 1 | 9 | 28 | 16 | 85 |
| Entries containing intentionally performed miracles ("IP" for short) | 14 | 1 | 1 | 5 | 1 | 0 | 2 | 2 | 0 | 12 |
| Percentage of IP among all entries | 22% | 20% | 100% | 17% | 3% | 0 | 14% | 2% | 0% | 6% |
| Percentage of IP among entries with miracles | 45% | 33% | 100% | 33% | 8% | 0 | 22% | 7% | 0% | 14% |

The percentage of narratives relating intentionally performed miracles as well as the diversity of such narratives appears to have significantly decreased in the *XBQNZ*. Almost all intentionally performed miracles find precedents in the *BQNZ*, except for *pubbenivāsānussati*. Nevertheless, many miracles in this category in the *BQNZ* are not found in records from after the Tang Dynasty in the *XBQNZ*. Records in this category touch only upon *pubbenivāsānussati*, effective divination, and supreme concentration during self-immolation.

Compared with various miraculous stories in the *BQNZ*, the miraculous responses related to death are more compelling in the *XBQNZ*. The table below shows the frequencies of such narratives in the *BQNZ* and *XBQNZ*.

Table 2 shows that the percentage of entries containing miracles related to death increased significantly in the Tang Dynasty. This increase maintained a steady trend, and the percentage of entries containing miracles related to death significantly surpassed the entries containing miracles with elements other than death during the Song Dynasty and reached its peak in the time of the Republic, when such entries constituted 94% of all entries containing miraculous narratives. Nevertheless, although smaller in number, miraculous narratives with elements other than death in the *XBQNZ* still cover almost all themes presented in the *BQNZ* and even add three new ones:

The first is about inviting rain by pious pray. As has been pointed out in Section 3, eminent nuns in the *BQNZ* were absent from several major areas where eminent monks showed their prowess, one of which is inviting rain. Of course, there is some difference between how the renounced ones invited rain in the *XBQNZ* and *GSZ*. Fotucheng and Shegong in the *GSZ* were said to have used spells and ordered the dragon king to make rain[39], while the eminent nun of Song just prayed for rain and her virtues brought rain as a spontaneous response from the nature. The latter put more emphasis on the miraculous response instead of wielding one's own miraculous power.

**Table 2.** Entries of miracles about death and those involving other elements.

| | BQNZ | XBQNZ | | | | | | | | | |
| | | Six Dynasties | Sui | Tang and Five Dynasties | Song | Yuan | Ming | Qing | The Republic | Total |
|---|---|---|---|---|---|---|---|---|---|---|
| Number of all entries | 65 | 5 | 1 | 30 | 29 | 8 | 14 | 86 | 27 | 200 |
| Entries containing miraculous narratives | 31 | 3 | 1 | 15 | 12 | 1 | 9 | 28 | 16 | 85 |
| Entries containing miracles involving death ("D" for short) | 8 | 1 | 0 | 7 | 9 | 1 | 7 | 25 | 15 | 65 |
| Entries containing miracles with elements other than death ("M" for short) | 26 | 3 | 1 | 10 | 5 | 0 | 5 | 6 | 3 | 32 |
| Percentage of D among all entries | 12% | 20% | 0 | 23% | 31% | 13% | 50% | 29% | 56% | 33% |
| Percentage of M among all entries | 40% | 60% | 100% | 33% | 17% | 0 | 36% | 7% | 11% | 16% |
| Percentage of D among entries containing miracles | 26% | 33% | 0 | 47% | 75% | 100% | 78% | 89% | 94% | 77% |
| Percentage of M among entries containing miracles | 84% | 100% | 100% | 67% | 42% | 0 | 56% | 21% | 19% | 38% |

The second is self-sacrifice other than self-immolation, where eminent nuns sacrificed their lives or parts of their bodies for the benefit of the temple or other people. The third pertains to the afterlife of an eminent nun of the Qing Dynasty, Lianzhen.

In summary, entries of "intentionally performed miracles" are far less diverse among miraculous narratives in the *XBQNZ* than in the *BQNZ*. Moreover, most miracles in later records of the *XBQNZ* are related to death. These accounts have a large number of similarities and seem to have forged a new indigenous repertoire.

According to Eliade, the history of religions is constituted by a large number of instances of *hierophany*, the act of manifestation of the sacred (Eliade 1959, p. 11). The miracles here can also be viewed as *hierophany*. The greater the extent to which monastics are revered and viewed as holy, the more often and diversely they are related to the *hierophant*—miraculous experience in this context. The less diversified miraculous narratives of later periods—after the Tang Dynasty—in the *XBQNZ* seem to signify less *hierophany* among eminent nuns in these periods. Several tentative interpretations of this are provided below.

First, the less *hierophany* for the eminent nuns of late imperial China and the time of the Republic might suggest that these later eminent nuns did not enjoy the same esteem as their predecessors in the Southern and Northern Dynasties and Tang Dynasty. This seems to accord with other descriptions in the *BQNZ* and *XBQNZ* too.

In the entry on Daorong, Baochang commented that her efforts led to Buddhism being esteemed in the Jin Dynasty[40]. In the *XBQNZ*, Facheng of the Tang Dynasty is praised for spreading the Dharma by both her contemporary eminent monks and the compiler (Shi 1983, pp. 10–11). She is also recorded as having been respected by the emperor. Daohui of the Song Dynasty impressed the royal family by successfully inviting rain. Henceforth, she preached Dharma to women in the court (Ibid. p. 41). In entries from and before the Song Dynasty, many eminent nuns were respected by the upper class and even by the emperor himself.

We cannot find such reverence in miraculous narratives after the Song Dynasty. Furthermore, the fame of eminent nuns became limited to local areas. No nun has been recorded as associating with the upper class. The secularization tendency of Buddhist nuns and the decline in the status of them in late imperial China have been noted by many scholars (Zurndorfer 1999, pp. 102–3; Yi 2006; Grant 2009, pp. 1–16; Guo 2010, pp. 108–21;

Zhang 2010, pp. 66–68), and the scarcity of accounts of *hierophany* in the *XBQNZ* of these periods can be explained by this.

Second, the less *hierophany* among eminent nuns in later periods might have reflected the development and social status of Buddhism in general. Based on the biographies of eminent monks, Liu proposed that Chinese people might have paid more attention to the supernormal powers of Buddhism during the Southern and Northern Dynasties, the early Tang Dynasty, and the middle of the Ming Dynasty because of the historical and cultural background in China and the development of Buddhism itself (Liu 2011). Buddhism thrived in the above-mentioned periods (Chen 1964; Du 2005). Therefore, monks were associated with *hierophany* to a greater extent when Buddhism thrived. His findings align with the research here on the miraculous narratives of eminent nuns. The frequencies of "intentionally performed miracles" as a percentage of all miraculous narratives in these three periods are significantly greater than in the other periods. In this sense, the less frequent *hierophany* of eminent nuns in later periods in the *XBQNZ* may be related to the decline of Buddhism in general. Nuns themselves do not need to bear full responsibility for it.

Third, the fewer instances of *hierophany* might have resulted from special social circumstances in particular times.

For example, in the Song Dynasty, Zen Buddhism flourished. Accordingly, almost half of the entries from the Song Dynasty narrate the Zen wit of eminent nuns. The miraculous experience of eminent nuns might have been eclipsed by their Zen wit in the eyes of the contemporary compilers; therefore, fewer miraculous narratives were recorded.

Another case in point is the Republican times. In records from this period, eminent nuns were related with the least *hierophany* of any period, where this might have been influenced by social circumstances.

In the time of the Republic, Buddhism in China encountered a new crisis. Modern science secularized mysterious phenomena (Yu 2007) and made miraculous narratives less attractive. This led to criticisms of Buddhism as superstition. The "smashing superstition" campaign in the 1920s made this issue more prominent. Under these circumstances, it is understandable that the *XBQNZ* seems to place less emphasis on miracles while focusing on the resilience, perseverance, and grit of eminent nuns in pursuing the Dharma and spreading it, their educational background, and their social engagements (Zhang 2005; DeVido 2015). Nevertheless, political turmoil and the chaos caused by wars affected the Chinese in the turbulent time of the Republic. People still had a psychological need for salvation and an urge for solutions to earthly problems, especially those concerning life and death. The miraculous stories, especially those concerning death, still had faithful audiences in such an environment. Therefore, narratives concerning death comprise most of the miraculous narratives from this period.

Besides the three above-mentioned interpretations, of course, there might also be others. For example, the growing importance placed on the study of Buddhist philosophy and metaphysics in later periods of Chinese Buddhism might also be one of the reasons for the statistical observation. Further investigation would surely shed more light on this topic.

## 5. Ideal Eminent Nuns through the Eyes of Male Monastics

A hagiography is mainly an expression of what the subject is supposed to be (Kieschnick 1997). The values of the compilers—the male monastics who were nurtured by traditional Chinese culture in this case—are reflected in their comments and constructions of their subjects. Therefore, is it possible that such agency on part of the compilers has undermined the contributions of Buddhist nuns to social engagement?

To answer this question, we start with a discussion of the compilers' attitudes toward eminent nuns. Sponberg has identified four attitudes in Buddhist texts toward women: soteriological inclusiveness, institutional androcentrism, ascetic misogyny, and soteriological androgyny (Sponberg 1992, pp. 7–29).

Soteriological inclusiveness is an important theme of the two collections considered here. In the preface of the *BQNZ*, Baochang acknowledged the outstanding religious achievements of eminent nuns. He also stated that his compilation of this biographical collection was motivated by his deploration of the fact that the aspirations and achievements of eminent nuns had never been documented before[41]. Similarly, in the preface of the *XBQNZ*, Zhenhua expressed his admiration for the accomplishments of the nuns and commented that men and women had the same Buddha nature and that the difference between men and women was simply a constructed idea. He also admitted that the ultimate religious achievements of eminent monks and nuns were the same. Thus, he lamented that no one had continued Baochang's work for more than one thousand years, which motivated him to embark on the arduous task of compiling a sequel to the *BQNZ* (Shi 2004). In all, both compilers showed great respect for the eminent nuns and thought they were equal to eminent monks with regard to their ultimate religious achievements and religious aspirations.

Soteriological androgyny, exemplified by Sponberg in Vajrayāna Buddhism, does not have a clear manifestation in the collections as few nuns engaged in Vajrayāna.

Ascetic misogyny does not find a manifestation in the collections considered here either. None of the miraculous narratives is related to debatable "gender transformation" to achieve enlightenment, and none of the nuns viewed their female bodies as impure or obstacles to enlightenment. On the contrary, Dumu jingang of the Ming Dynasty reportedly uttered verses about the irrelevance of gender regarding enlightenment[42].

Moreover, Faure (2003, pp. 202–3) viewed self-immolation by nuns as an extreme case of self-denial of the female gender. However, self-immolation was practiced by both monks and nuns in medieval and pre-modern China. According to collections on the lives of eminent monks, more than 50 reportedly attempted or committed self-immolation (Jan 1965). In this sense, self-immolation does not reveal a gender-specific focus but is rather related to selflessness and mercy regardless of gender.

However, in two entries from the Ming Dynasty that do not relate miracles, nuns have been praised for their "manly manners", a slight hint of gender discrimination. In addition, both the *BQNZ* and the *XBQNZ* record how eminent nuns resisted the sexual advances of men through miracles. In one entry from the Qing Dynasty (not relating a miracle), a nun tanned her skin to avoid male attention. Of course, the concern with female chastity reflects androcentric values. Nevertheless, none of the compilers faults the nuns for such incidents simply for having a female body. Even in the case of the nun who had tanned her skin, the compiler commented that she did it for convenience (to practice Dharma). Therefore, such narratives are not suitable examples of misogyny. Of course, the male villains here cannot represent all men; therefore, they are not suitable examples of misandry either.

Institutional androcentrism is, however, manifested in the collections. For example, in the *XBQNZ*, the miraculous narratives concerning the nuns Chifa and Huiren state that they were visited by the incarnation of Samantabhadra and their experience impressed Master Yixing. This narrative is an abbreviated version, and the inscription for two eminent nuns of the Tang Dynasty in the Tianxing nunnery of Changzhou[43] provides more details:

The two sisters later visited a famous master, Zhaoji[44], and claimed that his understanding was not deep enough. The master's disciples became angry at the "arrogance" of the two nuns and asked Master Yixing to censure them. However, after discussing Dharma with the two sisters, Yixing was impressed by their profound understanding and reported their eminence to the emperor. The emperor made a great offering to them and even built a nunnery for them.

Heirman (2010) has noted that the leading early Tang masters emphasized the subordination of Buddhist nuns. Such an emphasis seems to be part of a long tradition among Chinese monastics. In the *XBQNZ*, the above-mentioned narrative seemed to be adapted to ensure female subordination. The respect they received from the emperor and the nuns' contribution to spreading Buddhism were thus neglected.

Zhenhua does not identify his sources for the *XBQNZ*. From the postscript of this collection, we know that his main sources are classic books and anecdotes (Shi 1983, p. 151). The compilers of the series of hagiography collections of Chinese eminent monks were believed to have copied the text fully—or with minor additions or deletions—from sources available to them (Kieschnick 1997, p. 10). Zhenhua had probably followed this tradition. Therefore, such an abbreviation might have been done by Zhenhua himself or the earlier recorder, although we are not sure if he was a monastic. In this sense, this abbreviation might have reflected the institutional androcentrism in the Saṅgha or androcentric inclination of Chinese lay literati.

Actually, gender discrimination in Chinese culture does seem to have influenced the values of the compilers.

As mentioned above, there is no miracle related to eminent nuns serving as imperial advisors in the *BQNZ*, as is the case with their male counterparts in the *GSZ*. The office of the imperial advisor offered an important opportunity to lobby for Buddhism. Were eminent nuns from the Six Dynasties really excluded from this office?

According to the *BQNZ*, Miaoyin had influence over Emperor Xiaowu of the Jin Dynasty, even on critical political issues such the appointment of the governor of a province. However, Miaoyin's political involvements are not associated with any miracle in the *BQNZ*, which suggests that she was viewed by society or the compiler as less holy than those engaged in self-cultivation.

Moreover, in the Jin Dynasty, an unnamed nun used miraculous signs to warn a high official, Huanwen, to abandon his plans to usurp the throne[45]. In the Liu Song Dynasty, the nuns Fajing and Tanlan were involved in a conspiracy against the state and used their transportation privileges as nuns to send secret messages[46]. In the Northern Dynasties, many nuns, who used to be court women, still had an active role in politics after having gone forth (Miao 2011; Shi and Chen 2012; Bai 2016b). For example, in the Northern Wei Dynasty, Ciqing nurtured and protected two emperors (Zhou 2016). It is understandable to exclude Fajing and Tanlan, two political failures, from the hagiographic collections. Yet, the miracle of the unknown nun and Ciqing's contribution are also neglected by both Baochang and Zhenhua, for reasons not known.

Faure (2003, p. 214) suggests that when women wielded too much political power in ancient China, they were demonized because of their usurpation of "legitimate" male power. The ideal eminent nuns in Baochang's and Zhenhua's views might also have been apolitical and otherworldly (Wang 2021). Then, the lack of frequency of *hierophany* regarding nuns' involvement in politics might have resulted from the negative attitudes toward this practice of the male compilers, who were nurtured by the androcentric Chinese culture.

In summary, the two compilers have clearly confirmed the equality between eminent nuns and monks as far as their ultimate religious achievements and religious aspirations are concerned. Moreover, seen from the miraculous narratives, the two collections do not appear to discriminate based on gender and sex in general. However, seen from some miraculous narratives, institutional androcentrism and gender discrimination in Chinese culture appear to have constrained the compilers from presenting a comprehensive picture of the social engagements of eminent nuns. Therefore, the historical contributions of eminent nuns to spreading Dharma might have been underestimated.

## 6. Concluding Remarks

In this paper, the author first introduced miraculous narratives in the *BQNZ* and *XBQNZ*, and then provided a comparative examination based on the relevant narratives in the *BQNZ*, *XBQNZ*, and *GSZ*.

First, by comparing the miraculous narratives in the *BQNZ* and the *GSZ*, the author noted two apparent differences between the portrayals of nuns and monks. First, many foreign monks are reported to have had mysterious powers in the *GSZ*, while the *BQNZ* does not allocate any space to foreign nuns despite their apparent eminence. The *BQNZ*'s

silence on the eminence of foreign nuns, in sharp contrast to the special attention and glory afforded to foreign monks in the *GSZ*, might suggest androcentrism in both the Saṅgha and Chinese society. Second, the miracles of eminent nuns in the *BQNZ* seem to be more limited than those of their male contemporaries in the *GSZ*. Eminent nuns did not assume some important roles that eminent monks had frequently taken, such as healers and imperial advisors. Seen from these records, eminent nuns in the Six Dynasties did not engage in politics. Accordingly, their agency to spread Dharma might also have been limited.

Second, by scrutinizing the differences between miraculous narratives in the *BQNZ* and the *XBQNZ*, the author found that the entries containing "intentionally performed miracles" in the *BQNZ* exceeded those in the *XBQNZ* in terms of the percentage of all entries and diversity. Moreover, most miracles in later records of the *XBQNZ* are related to death. These two findings seem to signify less *hierophany* related to eminent nuns in later periods. This, in turn, might point to the lower esteem in which the eminent nuns of late imperial and pre-modern China were held. The author suggests that this might also be related to the development and social status of Buddhism in general. Moreover, the lower frequency of *hierophany* in the times of the Song Dynasty and Republic might have stemmed from special social circumstances.

Finally, the author addressed the question of whether eminent nuns in history actually primarily focused on self-cultivation and had limited agency in the context of social engagements. Seen from the miraculous narratives in them, the author suggests that the androcentric inclination, conscious or unconscious, of the male compilers of the *BQNZ* and *XBQNZ*, or of the compilers of the sources that the two collections rely on, might have led to an undermined portrayal of the prominence of nuns in spreading Buddhism. Such an androcentric bias is reflected in their selective use and adaptation of the materials.

Buddhist history is usually viewed as a male-centric history because men took full charge of compiling, distributing, and canonizing most of the primary sources of Buddhism. Buddhist nuns in general have held a marginal place in this context in the eyes of both male scholars and Buddhist monastics. Nevertheless, eminent Chinese Buddhist nuns have impressed male compilers and left behind a precious religious heritage despite patriarchal hegemonies, as evidenced from the *BQNZ* and the *XBQNZ*. Surely, there are more to be discovered. The achievements of these eminent Chinese nuns should be emphasized to enrich the history of Buddhism, impart precious knowledge on Buddhist women in history, encourage a more active role for Buddhist women in contemporary society, and promote the reformation of Buddhist institutions.

**Funding:** This research received no external funding.

**Institutional Review Board Statement:** Not applicable.

**Informed Consent Statement:** Not applicable.

**Data Availability Statement:** CBETA online: https://cbetaonline.cn/zh/ (accessed on 18 April 2023).

**Conflicts of Interest:** The author declares no conflict of interest.

## Glossary

| | |
|---|---|
| Aunt Kong | 空姑 |
| Baoxian | 寶賢 |
| Bendao | 本道 |
| Benlian | 本蓮 |
| Benyin | 本印 |
| Changjing | 常淨 |
| Changzhou | 常州 |
| Chaochen | 超琛 |
| Chaoyin | 潮音 |

| | |
|---|---|
| Chengci | 成慈 |
| Chengjing | 成静 |
| Chifa | 持法 |
| Ciqing | 慈慶 |
| Daogan | 道干 |
| Daohui | 道輝 |
| Daoji | 道跡 |
| Daoshou | 道壽 |
| Daoqiong | 道瓊 |
| Daorong | 道容 |
| Daowu | 道悟 |
| Daozong | 道綜 |
| Daizong | 代宗 |
| Dawu | 大悟 |
| Decheng | 德成 |
| Dele | 德樂 |
| Dengling | 等齡 |
| Demi | 德密 |
| Dumu jingang | 獨目金剛 |
| Emperor Jianwen | 簡文帝 |
| Emperor Xiaowu | 孝武帝 |
| Emperor Ming of the Jin Dynasty | 晉明帝 |
| Fabian | 法辯 |
| Facheng | 法澄 |
| Fahui | 法惠 |
| Fajing | 法淨 |
| Fakong | 法空 |
| Farun | 法润 |
| Fasheng | 法盛 |
| Faxuan | 法宣 |
| Faxiang | 法相 |
| Faxin | 法信 |
| Fayuan | 法緣 |
| Foqi | 佛琦 |
| Fotucheng | 佛圖澄 |
| Foyin | 佛音 |
| *gantong* | 感通 |
| *ganying* | 感應 |
| Gaofeng | 杲峯 |
| Gongdeshan | 功德山 |
| Guangjing | 光靜 |
| Guoren | 果仁 |
| Hongyuan | 宏源 |
| Huangxin | 黄心 |
| Huanwen | 桓溫 |
| Hui'an | 慧安 |
| Huimu | 慧木 |
| Huiqiong | 慧瓊 |
| Huiren | 慧忍 |
| Huisheng | 惠勝 |
| Huixiu | 慧秀 |
| Huiyao | 慧耀 |
| Huiyu | 慧玉 |
| Huiyuan | 惠源 |
| Jinggui | 淨珪 |
| Jingcheng | 靜稱 |
| Jingjian | 淨檢 |
| Jingxiu | 淨秀 |

| | |
|---|---|
| Jingzhen | 淨真 |
| Jingzhou | 荆州 |
| Jixing | 寂性 |
| Langran | 朗然 |
| Lianzhen | 蓮贞 |
| Liaozheng | 了证 |
| Lingyi | 灵一 |
| Lingzong | 令宗 |
| Lvzong | 律宗 |
| *MahāprajñāpāramitāŚāstra* | 《大智度論》 |
| Miaocheng | 妙成 |
| Miaokong | 妙空 |
| Miaoyin | 妙音 |
| Minggan | 明感 |
| Mingheng | 明恒 |
| Mount Wutai | 五臺山 |
| Nengfeng | 能奉 |
| Nengkai | 能開 |
| nun Hunshan | 混山尼 |
| nun Dong | 董尼 |
| nun Feng | 馮尼 |
| Pugui | 普贵 |
| Puzhao | 普照 |
| Qingyue | 清月 |
| Rujue | 如覺 |
| Ruzhan | 如湛 |
| Ruzhi | 如智 |
| Saṅghavarman | 僧伽跋摩 |
| Sengduan | 僧端 |
| Sengfa | 僧法 |
| Sengguo | 僧果 |
| Sengjing | 僧敬 |
| Shanmiao | 善妙 |
| Shegong | 涉公 |
| *sheli* | 舍利 |
| Shengdao | 聖道 |
| Shengping period | 昇平 |
| *shenyi* | 神異 |
| Shi Baochang | 釋寶唱 |
| Shi Zhenhua | 釋震華 |
| Suiqin | 遂钦 |
| *Sukhāvatīvyūha Sūtra* | 《阿彌陀經》 |
| Suwen | 素文 |
| Taiji hall | 太極殿 |
| Tanhui | 曇暉 |
| Tanjian | 曇簡 |
| Tanlan | 曇覽 |
| Tanyong | 曇勇 |
| The Chen Dynasty | 陳朝 |
| The Chinese Buddhist Canon | 中華大藏經 |
| The East Jin Dynasty | 東晉 |
| The Lanruo nunnery | 蘭若庵 |
| The Liang Dynasty | 梁 |
| The Liu Song Dynasty | 劉宋 |
| The Ming Dynasty | 明朝 |
| The Northern Wei Dynasty | 北魏 |
| The Qi Dynasty | 齊 |
| The Qing Dynasty | 清朝 |

| | |
|---|---|
| The Republican times | 民國 |
| The Rebellion of An and Shi | 安史之亂 |
| The Tang Dynasty | 唐朝 |
| The Song Dynasty | 宋朝 |
| The Sui Dynasty | 隋朝 |
| The Yuan Dynasty | 元朝 |
| Tianjian period | 天監 |
| Tianxing | 天興 |
| Tongdao | 通道 |
| Wendi of the Sui Dynasty | 隋文帝 |
| Wuwei | 無為 |
| Wuxing | 悟性 |
| Xinggang | 行剛 |
| Xingxuan | 行玄 |
| Xinxiang | 信相 |
| Xinzhong | 心忠 |
| Xuanzang | 玄奘 |
| Xuanzao | 玄藻 |
| *Yogācāra-bhūmi* | 《瑜伽師地論》 |
| Yingen | 印根 |
| Yinxin | 印心 |
| Yixing | 一行 |
| Yuanji | 元機 |
| Zhaoji | 照寂 |
| Zhenru | 真如 |
| *zhiguai xiaoshuo* | 志怪小說 |
| Zhisheng | 智勝 |
| Zhixian | 智賢 (of the Jin Dynasty) |
| Zhixian | 智仙 (of the Sui Dynasty) |
| Zhiyue | 直月 |
| Zhizang | 智藏 |

## Notes

1  CBETA 2023.Q1, B35, no. 194, p. 762.

2  「謂六神通：一者神足通證，二者天耳通證，三者知他心通證，四者宿命通證，五者天眼通證，六者漏盡通證。」(CBETA 2022. Q4, T01, no. 1, pp. 58a24–26)「如意足智通，天耳智通，他心智通，宿命智通，生死智通，漏盡智通」) (CBETA 2022.Q4, T01, no. 26, pp. 553b04–554b12).

3  Georgieva (1996, pp. 59–60) categorized the miracles into three categories. I combined her first category of "miraculous response which happens in in answer to a prayer for help and serves to save or protect them" and second category of "unexpected events resulting from faith and practice of the main character" or "solicited miracles" into "miraculous responses" as I think that these "unexpected events" are also miraculous responses evoked by the faithful from nature.

4  CBETA 2022.Q4, T50, no. 2063, p. 936.

5  Ibid. p. 935.

6  Ibid. p. 941.

7  Ibid. pp. 943–44.

8  Ibid. p. 945.

9  Ibid. pp. 939–40.

10  Ibid. p. 940.

11  Ibid. pp. 940–41.

12  Ibid. p. 939. There are three more nuns who performed self-immolation in the *BQNZ*, namely, Tanjian, Jinggui, and Tanyong (Ibid. pp. 943–44). I do not include these narratives in the discussion because their self-immolation is just briefly mentioned without a description of a supernormal experience indicating their superb meditation skills.

13  Ibid. p. 946.

14  Ibid. p. 935.

15  Ibid. p. 936.

16    Ibid. p. 937.
17    Ibid. p. 939.
18    Ibid. p. 938.
19    Ibid. p. 943.
20    Ibid. p. 945.
21    Ibid. p. 935.
22    Ibid. p. 939.
23    Ibid. p. 935.
24    Ibid. p. 936.
25    Ibid. p. 939.
26    Ibid. p. 935.
27    Ibid. p. 940.
28    Ibid. p. 938.
29    Ibid. p. 942.
30    Ibid. p. 948.
31    Ibid. p. 946.
32    Ibid. p.935. *Karma* in Sanskrit, *kamma* in Pāli, or 羯摩 in Chinese, in a broad sense, refers to ceremonies and etiquette. Normally, it refers to the etiquette for receiving precepts, repentance, barriers, etc.
33    Ibid. p. 940.
34    Ibid. p. 938.
35    Ibid. pp. 939, 941.
36    CBETA 2023.Q1, T50, no. 2059, p. 342.
37    " 我悲常寂" (Shi 1983, pp. 125–26).
38    White birds ( 白雀) were viewed as an auspicious omen by the ancient Chinese.
39    CBETA 2023.Q1, T50, no. 2059, pp. 385, 389.
40    " 以後晉顯尚佛道容之力也。" CBETA 2022.Q4, T50, no. 2063, p. 936.
41    " 志業未集乎方冊。每懷慨歎。" CBETA 2023.Q1, T50, no. 2063, p. 934.
42    " 男女何须辨假真，观音出果何人。皮囊脱尽挥无用，试问男身是女身。" (Shi 1983, p. 54).
43    The full inscription is found in Noriko Katsuura ( 勝浦令子). 1993. " 法華滅罪之寺と洛陽安国寺法華道場: 尼と尼寺の日唐比較研究の課題" 《史論》46: 1–18.
44    Maybe it should be Puji 普寂, a famous Chan master in Tang.
45    See *Fayuan zhulin* 《法苑珠林》CBETA 2022.Q4, T53, no. 2122, pp. 545a22–29.
46    See "the Biography of Fanye", volume 69, *the Book of Song* 《宋書. 範曄傳》.

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
