# Peer review of "The Miraculous Narratives in The Biographies of Eminent Nuns and The Further Biographies of Eminent Nuns"

_religions, doi:10.3390/rel14050565_

Round 1

Reviewer 1 Report

For a journal article, it should have a coherent central argument and a clearer structure rather than cram in every available reference. The overall presentation needs to be much improved, otherwise it is very difficult to read. Footnotes need to be collated properly and use either Harvard or Chicago style (but not both). Chinese characters can be moved to a Glossary or inserted only for the first instance.  Please reduce the abbreviations of book titles as it is such a put-off.    

Reviewer 2 Report

Thank you for the opportunity to read this excellent and interesting article. I congratulate the author(s) for their methodical work. The author(s) drew on miracle accounts in The Biographies of Eminent Nuns and The Further Biographies of Eminent Nuns to situate the status of bhikṣuṇīs in the eyes of their male monastic chroniclers, thereby reflecting the peripheral roles of these eminent nuns in Chinese society. The author(s) conducted a meticulous analysis of these hagiographies and observed a significant decline in both the quantity and diversity of accounts in intentionally performing miracles since the Sui Dynasty, except for a brief surge in the Ming Dynasty. These narratives, when present, were typically centered around the theme of death. Together with their discovery of the absence of any accounts of foreign nuns, the author(s) concluded that there was an inherent male-centric bias during the time of the compilation of these biographies.

This study ambitiously sets out to fill in some gaps in the study of Chinese bhikṣuṇīs from the Six Dynasties to the Republic era (spanning over 1,700 years).  The author(s) used the Biographies as the primary source to examine gender disparities between the portrayal of monks and nuns. Moreover, they analyzed changes in the miraculous narratives to infer shifts in societal attitudes over time. The first objective was largely accomplished. Although section 5 about the perception of monk compilers was well-structured, I suggest adding other perspectives that provide a more balanced view. For example, Baochang embarked on this arduous task of compiling the biographical accounts because he “deplored the fact that … the achievement of the aspirations” of the Buddhist nuns were not documented (Tsai 1981: 3). While it is not difficult to find indications of gender bias in Chinese society, evidence that suggests otherwise holds greater significance.

I have some reservation about this article’s conclusion of how nuns were perceived by monk compilers being drawn solely from the analysis of miraculous narratives in the Biographies. Though these narratives comprise over 40% of the Biographies, relying solely on them to make definitive conclusions is a cause for concern. For example, how does the observation that a decline in hierophany and increase in death miracle narratives indicate that there was a lower regard for eminent nuns in later periods (p. 19)? An article that the author(s) may consider is Hsieh, Ding-hwa. 2000. “Buddhist Nuns in Sung China (960-1279).” Journal of Song-Yuan Studies 30: 63-96. Although Hsieh did not base her research on the Biographies, her study reveals that during the early Song period, nuns were highly respected, and some prominent nuns were recorded by the court historian (Hsieh 2000: 83). Given that the miraculous narratives account for about 43% of the records in the Biographies, it may be more judicious to qualify assertions made. Also, the author(s) aptly noted the growing importance placed on the study of Buddhist philosophy and metaphysics over the years as another explanation for statistical trends observed. To avoid appearing like hasty generalizations, I recommend to temper the language of broader conclusions by acknowledging that they are founded on the study of 115 hagiographic accounts spanning over 1,700 years.

I find this article well-structured and the tables informative. I would like to recommend adding CBETA citations to section 4, as was so well-done in section 3. Is there also a missing in-text citation in lines 49-50? A footnote reference can be added to line 648 when referring to “many scholars” and line 698 on “verses about the irrelevance of gender regarding enlightenment.” In addition, the reference to “engaged in karma” in line 282, “the omens during the karma” in line 533, and “from the nature” in footnote 10 need further elaboration.

Overall, this manuscript carried an informative title and an excellent abstract. The author(s) clearly laid out the research questions and the findings are relevant to the focus on ‘biographies of bhikkunis, priestesses, female religious practitioners, performers, Chinese vegetarian nuns (zhaigu), etc.’ in the Special Issue on “Re-staging the Periphery as the Center: Women Communities in East Asian Religions.”

Reviewer 3 Report

The article is well structured and gives a good overview of miraculous stories of nuns in BQNZ and XBQNZ in comparison, revealing aspects that have not been fully studied before. There are a few things that may be improved though, as listed below. Among them #2 is the most important thing that needs to be addressed in revision.

#1. Page 4, line 123-126: XBQNZ “….composed in pre-modern time, also provides abundant material on eminent Buddhist nuns. Compiled by the Buddhist monk Shi Zhenhua (釋震華, 190847 CE) the XBQNZ is part traditional hagiography and part modern biography.”

“Pre-modern” and “modern” can be changed to more precise terms here. What “pre-modern time”? If it was considered as compiled in “pre-modern time”, why it also has “modern biography”?

#2. Page 4, after line 133-134:

It may be better to add a concise summary here of the validity of XBQNZ for the purpose of historical comparison in the current paper: As it covers such a long timespan, what were the sources of the stories in it? What’s the current scholarly consensus on their usefulness in studying nuns living a few hundreds to more than a thousand years before the compiler’s lifetime?  

This is particularly important considering the main arguments of this paper, that “later eminent nuns did not enjoy the same esteem as their predecessors in the Southern and Northern Dynasties” (p.16, line 629), and “…suggests a decline in the status of eminent nuns in the Song Dynasty” (p.16, line 644), etc. For BQNZ, which was compiled during the Southern and Northern Dynasties, we can be relatively confident that it at least reflects some aspects of its time period; but as XBQNZ was compiled in the 20th century, does it really reflect the social attitudes to nuns of the Six dynasties, Tang, Song, Yuan, Ming and Qing, or does it just reflect the attitude of the compiler who lived in the early 20th century? There needs to include some current academic discussion to clarify this, e.g., how comprehensive was Shi Zhenhua in collecting the stories of nuns? Are there possibilities that there are miraculous stories in other sources that were overlooked by him, which may portray nuns in those dynasties differently? The paper does touch a bit on the role of compilers in Section 5 (p.17), but does not give an adequate discussion of the usefulness of XBQNZ in terms of supporting a historical comparison.    

#3. Page 8-9, regarding the lack of miraculous stories of foreign nuns:

This is an interesting point, however, I wonder if this can really logically prove the point of “clear androcentrism” in the Sangha and society as the article attempts to do. Are there statistics of the percentage of accounts of foreign monks performing miracles against the total number of accounts of foreign monks mentioned in sources of the time period? Say, if there are 100 mentioning of foreign monks in total (from both the eminent monks’ biographies and other textual sources), and 10 among them are related to miracles, while there are only 5 mentioning of foreign nuns in total (from both the eminent nuns’ biographies and other textual sources), and none among them were about miracles, then it cannot prove the point that there was the intentional overlooking of foreign nuns performing miracles, but just that there might not be that many foreign nuns arrived in China, or foreign nuns in general did not catch much attention.

Nonetheless, I think this lack of miraculous narratives of foreign nuns is definitely an interesting thing to be mentioned, but it may be better to have a more careful investigation of the possible specific reasons for it, instead of just attributing it to a general “androcentrism”, which existed in any premodern (and modern) societies, as we all know.  

Round 2

Reviewer 1 Report

I wonder if all the Chinese characters are necessary and whether having them in the main text enhances the argument?  It could just have a glossary at the end and footnotes should also be used more effectively. 

Author Response

Thanks  a lot for providing the thoughtful and constructive comments!

I have move all the Chinese characters to the Glossary as suggested, leaving only Chinese names of the three biographical collections for the first instance. 

I also combined several footnotes to make them more effective.